# High-depth spatial transcriptome analysis by photo-isolation chemistry

Mizuki Honda [1,2,4], Shinya Oki [1,2,4 ✉], Ryuichi Kimura[2], Akihito Harada[3], Kazumitsu Maehara[3], Kaori Tanaka[3], Chikara Meno[1] & Yasuyuki Ohkawa [3 ✉]

In multicellular organisms, expression profiling in spatially defined regions is crucial to elucidate cell interactions and functions. Here, we establish a transcriptome profiling method coupled with photo-isolation chemistry (PIC) that allows the determination of expression profiles specifically from photo-irradiated regions of interest. PIC uses photo-caged oligodeoxynucleotides for in situ reverse transcription. PIC transcriptome analysis detects genes specifically expressed in small distinct areas of the mouse embryo. Photo-irradiation of single cells demonstrated that approximately 8,000 genes were detected with $7 \times 10^4$ unique read counts. Furthermore, PIC transcriptome analysis is applicable to the subcellular and subnuclear microstructures (stress granules and nuclear speckles, respectively), where hundreds of genes can be detected as being specifically localised. The spatial density of the read counts is higher than 100 per square micrometre. Thus, PIC enables high-depth transcriptome profiles to be determined from limited regions up to subcellular and subnuclear resolutions.

[1] Department of Developmental Biology, Graduate School of Medical Sciences, Kyushu University, Fukuoka, Japan. [2] Department of Drug Discovery Medicine, Kyoto University Graduate School of Medicine, Kyoto, Japan. [3] Division of Transcriptomics, Medical Institute of Bioregulation, Kyushu University, Fukuoka, Japan. [4]These authors contributed equally: Mizuki Honda, Shinya Oki. ✉email: oki.shinya.3w@kyoto-u.ac.jp; yohkawa@bioreg.kyushu-u.ac.jp

  

Each cell type in a multicellular organism is characterised by its gene expression profile, which is partly defined by its spatial context. The characterisation of whole organs and tissues has recently been advanced by technologies enabling genome-wide expression analysis, as represented by RNA-seq[1–3]. Properties of individual cells can be studied using single-cell RNA-seq by isolating individual cells from dissociated tissues[4–10]. Expression profiling with respect to spatial information is crucial to determine the characteristics of cells that are controlled, at least in part, by spatial interactions.

Expression patterns of individual or small numbers of genes are classically determined by in situ hybridisation (ISH) using riboprobes that are visualised by multi-colour staining[11–13]. The localisation of hundreds to thousands of transcripts can be analysed by seqFISH+ and MERFISH by using multiple riboprobe presets followed by sequential rounds of hybridisation[14,15]. Unbiased expression profiling associated with spatial information is enabled by spatial transcriptomics, Slide-seq, and HDST technologies that use multiplexed sequencing with positional barcodes[16–18], and FISSEQ and ExSeq technologies that conduct in situ sequencing[19,20]. Cells located in specific regions of interest (ROIs) can be isolated from specimens by laser-capture microdissection (LCM) and can then be analysed by RNA-seq[21]. The NanoString GeoMx Digital Spatial Profiler enables expression profiling of several hundred probe sets for light-exposed regions[22]. However, deep expression profiling of small numbers of cells or intracellular components remains technically challenging.

By taking advantage of the fact that light can change molecular properties with a resolution up to the diffraction limit, we here introduce photo-isolation chemistry (PIC), which enables transcriptional profiles of photo-irradiated cells alone to be determined. We demonstrate that PIC can provide detailed gene expression profiles for several dozen cells in small ROIs in mouse embryonic tissues. PIC identifies genes uniquely expressed in spatially distinct areas with a resolution up to subcellular and subnuclear levels.

## Results

**Establishment of PIC expression analysis**. To understand multicellular systems within a spatial context, we attempted to develop a gene expression profiling method for small areas with high spatial resolution. The method, termed photo-isolation chemistry (PIC), takes advantage of photo-caged oligodeoxynucleotides (caged ODNs) for the amplification of cDNAs in response to photo-irradiation (Fig. 1a). Experimentally, first-strand cDNAs are synthesised in situ by applying both the caged ODNs and reverse transcriptase onto the tissue sections (termed in situ RT)[23]. Optionally, regions of interest (ROIs) can be precisely defined by immunostaining with antibodies against regional markers. The caged moieties are then cleaved from the ODNs by specific wavelength photo-irradiation under a conventional fluorescence microscope. The whole specimen is then lysed with protease and cDNA:mRNA hybrids are extracted. Only the uncaged libraries can be amplified by CEL-seq2[24], a highly sensitive single-cell RNA-seq technology, by taking advantage of T7 promoter-driven linear amplification (termed in vitro transcription or IVT)[25]. In this manner, only gene expression from photo-irradiated ROIs is detected.

A critical issue in the development of PIC is the suppression of cDNA amplification from nonirradiated regions. The 6-Nitropiperonyloxymethyl deoxythymidine (NPOM-dT) was chosen to synthesise the caged ODNs because chain extension by DNA polymerases pauses at NPOM-dT sites in template strands, but resumes upon photo-irradiation of a wavelength around 365 nm[26]. DNA polymerase I, which is used for second-strand DNA synthesis in CEL-seq2, processes through single NPOM-dT sites irrespective of photo-irradiation[27,28]. To block DNA polymerase I read-through in PIC, we tried to insert multiple cages in the template strand (Fig. 1b). ODNs harbouring triplet NPOM-dTs were annealed with fluorescent dye-labelled primers (19 nt) and chain elongation by DNA polymerase I was examined with or without photo-irradiation. As a result, fully extended complementary strands (41 nt) were produced after photo-irradiation (352–402 nm for 15 min). In contrast, an extension was paused at the NPOM-dT triplet (24 nt) site in the absence of photo-irradiation, suggesting that repetitive NPOM-dT insertion effectively suppresses read-through of DNA polymerase I. On the basis of these results, we designed several caged ODNs and evaluated their signal-to-noise (S/N) ratios by comparing the amounts of amplified libraries with and without photo-irradiation (Fig. 1c). Total RNAs from NIH/3T3 cells (40 ng) were reverse-transcribed with caged ODNs and photo-irradiated or not under the fluorescence microscope (352–402 nm for 15 min). The cDNAs were then amplified by second-strand synthesis and IVT reactions. The resulting amplified RNAs (aRNAs) of housekeeping genes were quantified by RT-PCR to calculate the S/N ratio. The S/N ratio of noncaged ODNs was nearly equal to 1, indicating that 365 nm UV irradiation was minimally harmful to the integrity of nucleotides. A substantial amount of background was detected by insertion of an NPOM-dTs triplet at the 5′ terminal of the poly-T, with the S/N ratios ranging from $10^{0.96}$ (Gusb) to $10^{3.83}$ (Eef1a). The insertion of six NPOM-dTs in the adaptor sequence elevated the S/N ratios, ranging from $10^{1.90}$ (Gusb) to $10^{4.11}$ (Gapdh). The additional insertion of a single NPOM-dT into the poly-T sequence increased the S/N ratios, ranging from $10^{2.89}$ (Eef1a) to $10^{6.27}$ (Actb). One further NPOM-dT inserted into the poly-T sequence further suppressed the background, with the S/N ratios ranging from $10^{3.12}$ (Gusb) to $10^{7.13}$ (Gapdh). This last caged ODN was the most effective at suppressing background (Fig. 1d) and was therefore used for the experiments described below.

We also examined conditions to enhance the detection sensitivity of PIC. Hydrochloric acid (HCl) is used to enhance the efficiency of the in situ RT reaction[29], probably by removing proteins covering RNAs; therefore, the effect of HCl in the PIC platform was examined. GFP-expressing NIH/3T3 cells grown on coverslips (~5000 cells) were fixed and permeabilised in the presence or absence of HCl. In situ RT with noncaged ODNs (same sequences as Fig. 1d) was then performed. First-strand cDNA synthesis was detected without HCl treatment, but there was a 38-fold greater yield of cDNA with HCl treatment (qPCR for Gfp cDNA; Fig. 1e). HCl treatment also increased the amount of sequence library produced, but the length of cDNAs in the library was not affected (~200–500 bp; Fig. 1e). The optimal duration of photo-irradiation was also examined. GFP-expressing NIH/3T3 cells on coverslips were fixed and permeabilised with HCl, followed by in situ RT with caged ODNs. Coverslips were then irradiated with 340–380 nm light for varying times under the fluorescence microscope and cell lysates were then subjected to qPCR for Gfp cDNA. Gfp cDNA was detected at comparable levels after photo-irradiation for 15 and 30 min (Fig. 1f). This level was decreased by up to 1% following irradiation for 5 min, but no cDNAs were detected after irradiation for 1 min. Taking these optimizations together, the optimal PIC conditions were as follows: Specimens were permeabilised with HCl and heat-denatured; caged ODNs, as shown in Fig. 1d, were used for in situ RT; and 340–380 nm photo-irradiation for 15 min was used for uncaging. All of the experiments described below were performed using these conditions.

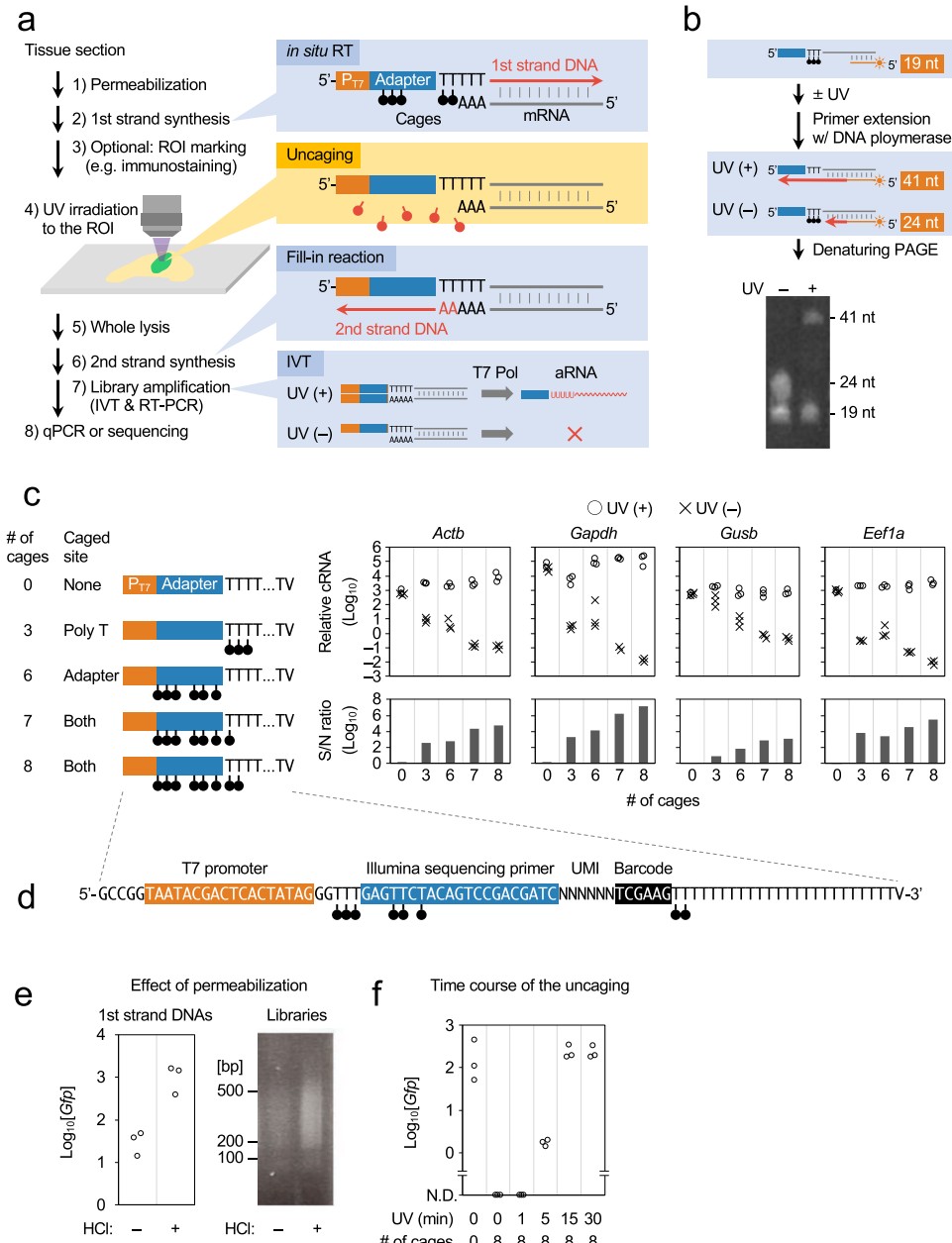

**Fig. 1 Establishment of the PIC expression profiling method. a** Schematic overview of expression profiling with PIC. $P_{T7}$, T7 promoter; T7 Pol, T7 polymerase. **b** Strategy (top) and the result (bottom) of primer extension experiments to suppress read-through of DNA polymerase I at the NPOM-caged dT sites (black circles) without UV irradiation. A representative image was shown out of three independent experiments. **c** Various caged ODNs were examined for RNA-seq library preparation with (circles) or without (crosses) photo-irradiation before quantifying the expression of *Actb*, *Gapdh*, *Gusb*, and *Eef1a* genes. **d** Sequence of the caged ODN showing the lowest background in (**c**). **e** Effect of HCl permeabilisation before in situ RT in GFP-expressing NIH/3T3 cells was evaluated by the yield of *Gfp* cDNA (left) and the size (right) of sequence libraries. A representative image was shown out of three independent experiments. **f** GFP-expressing NIH/3T3 cells were photo-irradiated for various times after in situ RT, and the yield of *Gfp* cDNA was quantified. Source Data is available as a Source Data file.

**Validation of PIC for ROI-specific profiling**. Suppression of the background from nonirradiated regions is crucial for ROI-specific expression profiling with PIC. To evaluate background levels, mixed-species cultures of human- and mouse-derived cell lines [T-47D (human) and *Gfp*-expressing NIH/3T3 (mouse)] were examined to detect species-specific expression after photo-irradiation (Fig. 2a and 3). T-47D and NIH/3T3 cells were separately aggregated and then mixed and adhered onto coverslips. They were then fixed, permeabilised, and in situ RT was performed. Both GFP-negative and -positive aggregates were UV-irradiated under the fluorescence microscope. The

objective lens was ×40, enabling irradiation of circular areas 750 μm in diameter. After irradiation, cell lysates were analysed by qPCR for species-specific gene expression. When human T-47D cells were photo-irradiated, human *GAPDH* was detected ($n = 3/4$), but mouse *Gapdh* and *Gfp* were not ($n = 0/4$ each; Fig. 2b, lane 5). In contrast, when *Gfp*-expressing NIH/3T3 cells were photo-irradiated, mouse *Gapdh* and *Gfp* were detected ($n = 4/4$ each), but human *GAPDH* was not ($n = 0/4$; Fig. 2b, lane 6), indicating that background from nonirradiated cells was below the detection limit in mixed culture experiments.

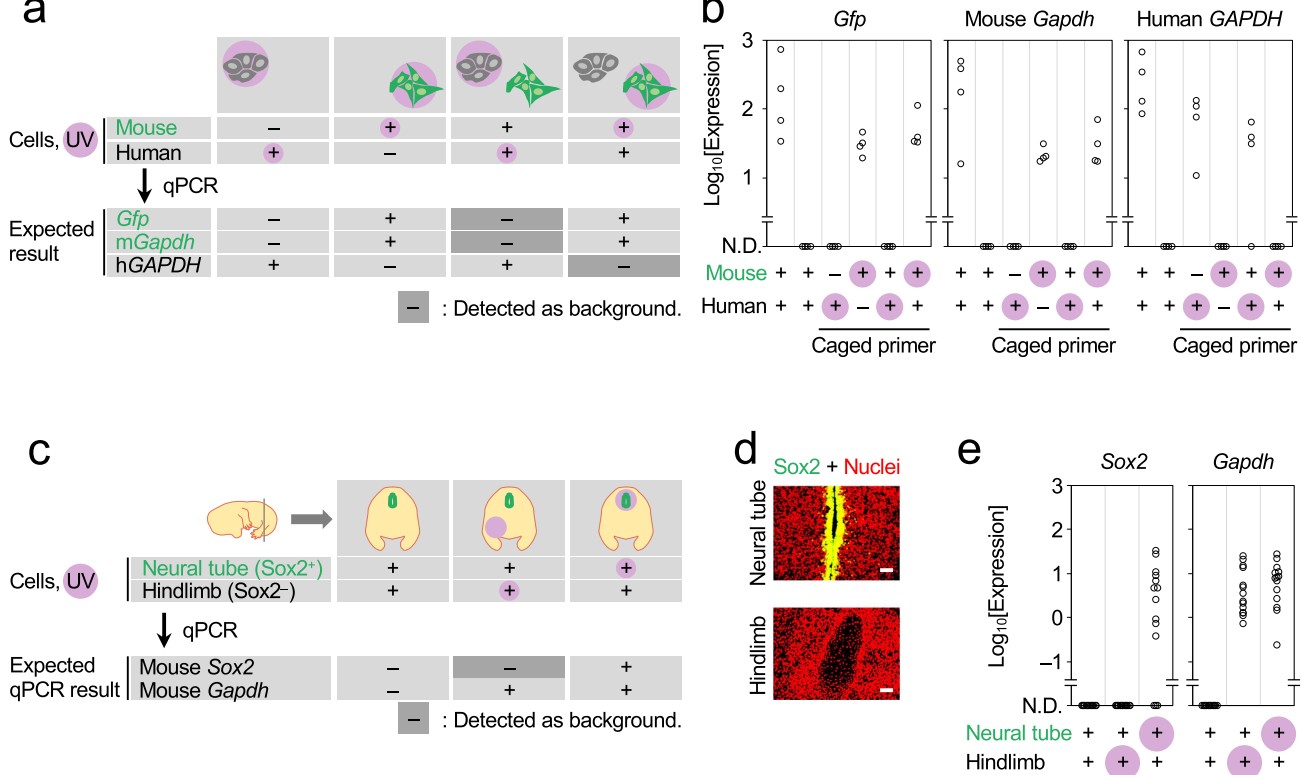

**Fig. 2 Validation of PIC for ROI-specific profiling. a–e** Experiments to evaluate the level of background from nonirradiated cells were performed with human–mouse mixed cultures (**a**, **b**) and E14.5 mouse embryos (**c–e**). After photo-irradiation (pink) of either human (T-47D, grey) or mouse (GFP-expressing NIH3T3, green) cells (**a**), the cDNAs of human (*GAPDH*) and mouse (*Gapdh* and *Gfp*) genes were examined (**b**). Mouse embryonic sections of Sox2-positive (neural tube, green) or -negative (hindlimb) cells (**c**, **d** a representative image was shown out of the replicates) were photo-irradiated and the cDNAs of *Sox2* and *Gapdh* genes were examined (**e**). Scale bars, 50 μm; red in D, nuclei. Source Data is available as a Source Data file.

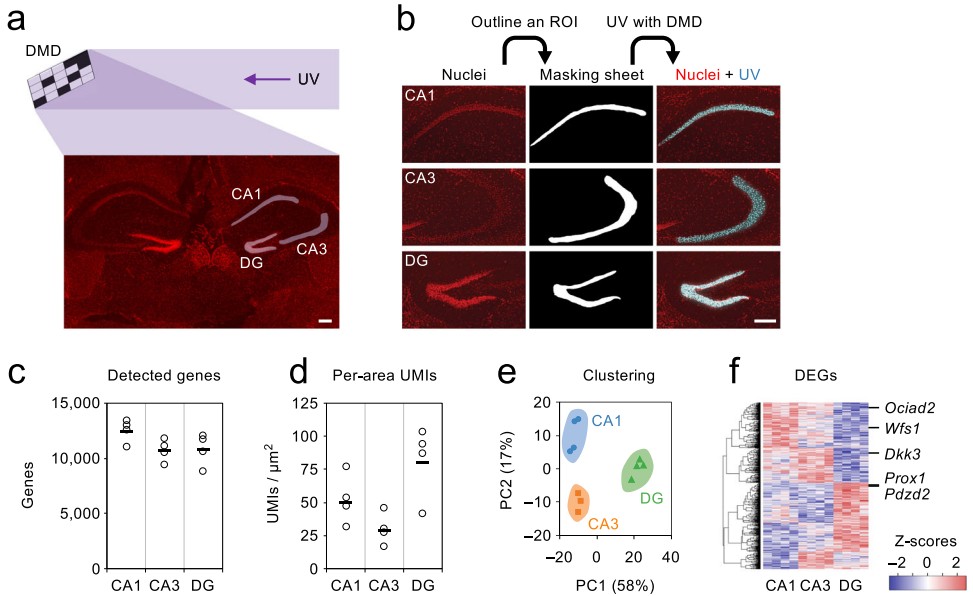

**Fig. 3 PIC RNA-seq for hippocampus under DMD illumination. a** Illustration of the experiments for DMD-assisted UV irradiation of CA1, CA3, or DG of mouse hippocampus. Samples were tested in four biological replicates. **b** Images of nuclei (red), resulting in masking sheets, and merged images of nuclei and DMD-assisted UV irradiation (blue) are shown. **c, d** Numbers of detected genes (**c**) and per-area UMIs (**d**) are shown, with the horizontal lines indicating the average of replicates. **e** Two-dimensional PCA for the expression profiles. **f** DEGs are clustered and shown in heat maps. Scale bars, 200 μm. Source Data is available as a Source Data file.

We next applied PIC to tissue sections to determine gene expression in photo-irradiated cells (Fig. 2c). Fresh frozen sections of embryonic day (E) 14.5 mouse embryos were fixed, permeabilised, and subjected to in situ RT. The sections were then immunostained for SOX2 to label the midline of the neural tube (Fig. 2d). Either SOX2-positive or -negative cells (neural tube or hindlimb, respectively; ×20 lens to irradiate circular areas 1.2 mm in diameter) were then UV-irradiated. After irradiation, tissue lysates were collected and analysed by qPCR for *Sox2* and ubiquitously expressed *Gapdh*. When the neural tube was photo-irradiated, both *Sox2* and *Gapdh* were detected ($n = 11/14$ and 14/14, respectively; Fig. 2e, lane 3). In contrast, when the hindlimb was photo-irradiated, *Gapdh* but not *Sox2* expression was detected ($n = 14$ each; Fig. 2e, lane 2). These findings demonstrated that background from nonirradiated cells was detected in neither cultured cells nor tissue sections.

**Sensitivity of PIC with qPCR**. To examine the detection sensitivity of PIC, varying numbers of cells were photo-irradiated for expression analysis. *Gfp*-expressing NIH/3T3 cells were sparsely inoculated onto coverslips (10,000 cells), followed by fixation, permeabilisation, in situ RT, and photo-irradiation. The number of irradiated cells was adjusted by changing the magnitude of the objective lens as follows: 2392–2766 cells (×2.5 lens), 596–860 cells (×5), 182–239 cells (×10), 53–66 cells (×20) and 17–25 cells (×40). After irradiation, the cell lysates were analysed by qPCR for *Gfp* and *Gapdh*. The smallest numbers of cells capable of detecting *Gfp* and *Gapdh* were 20 and 17, respectively (Supplementary Fig. 1a). The quantified expression levels increased with increasing cell numbers. cDNAs were not detected in the absence of photo-irradiation ($n = 0/4$).

Detection sensitivity was also evaluated in tissue sections. Frozen E14.5 mouse embryo sections were subjected to in situ RT and immunostaining for SOX2. Photo-irradiation of the neural tube was performed with a ×100 objective lens. The sizes of the photo-irradiated areas were adjusted using the fluorescent field diaphragm of the fluorescence microscope, which was changed in a stepwise manner as follows: φ 250 μm (~ 200 cells), φ 75 μm (~80 cells) or φ 16 μm (~ 10 cells) (Supplementary Fig. 1b). After irradiation, the tissue lysates were analysed by qPCR for *Sox2* and *Gapdh*. A photo-irradiation diameter of at least 16 μm could detect *Sox2* and *Gapdh* expression ($n = 2/16$ and 9/16, respectively; Supplementary Fig. 1c). A photo-irradiation diameter of at least 75 μm could detect the expression of both genes more frequently ($n = 5/16$ and 14/16, respectively). For a photo-irradiation diameter of 250 μm, *Sox2* was detected in half and *Gapdh* in all samples ($n = 8/16$ and 16/16, respectively). These results indicate that PIC can detect gene expression from 10 or more cells in tissue sections.

**PIC RNA-seq for tissue sections**. In the developing neural tube, multiple subtypes of interneurons and motor neurons are generated along the dorsoventral axis[30]. Such neural tube patterning is established by opposing morphogen gradients of BMP/WNT and SHH proteins from dorsal and ventral domains, respectively[31,32]. Although a number of genes have been identified to be expressed along the dorsoventral axis by histological methods such as ISH[33], genome-wide expression profiling of each domain remains challenging; this is because of the difficulty of isolating such small numbers of cells in a precise manner. We performed PIC to isolate expression profiles from small domains of the neural tube. Following on from our previous experiments (Supplementary Fig. 1b), E14.5 mouse sections were subjected to in situ RT and immunostaining for SOX2. Three distinct domains of the neural tube were independently photo-irradiated

(dorsolateral, mediomedial, and ventromedial sites; $n = 6, 4$, and 6, respectively, Supplementary Fig. 2a), using a ×100 objective lens with a fluorescence field diaphragm (irradiation diameter = 75 μm). As controls, we prepared nonirradiated samples ($n = 4$) and samples in which larger areas were irradiated, centred on the midline of the neural tube (φ 250 and 5000 μm using ×100 and ×5 lenses, respectively, with an open diaphragm; $n = 4$ each). The specimens were then lysed and libraries were amplified and sequenced to allow $1 \times 10^7$ reads per sample. On average, sequencing generated approximately 10 million reads per sample, irrespective of irradiation size ($1.0 \times 10^7$, $5.8 \times 10^6$, and $1.0 \times 10^7$ read for irradiations of φ 5000, 250 and 75 μm, Supplementary Fig. 2b). In comparison, the number of reads from nonirradiated samples was small ($4.3 \times 10^4$ reads), indicating that background reads were rarely included in the reads of photo-irradiated samples. In the photo-irradiated samples, more than half of the reads uniquely mapped to the mouse reference genome (Supplementary Fig. 2c). The mapability was lower than that of the original CEL-seq2 methods[24] (> 80%), which was potentially due to modifications such as fixation, in situ RT and proteinase lysis, and/or the lower quality of library synthesis affected by contamination of a large amount of proteins and genomic DNAs. The number of gene-assigned reads corrected by unique molecular identifiers (UMIs) was dependent on the size of the irradiated area ($4.0 \times 10^5$, $1.9 \times 10^5$, and $4.0 \times 10^4$ reads for φ 5000, 250 and 75 μm irradiations, respectively, Supplementary Fig. 2d). Remarkably, more than 10,000 protein-coding genes were detected, even in the smallest areas irradiated ($1.6 \times 10^4$, $1.4 \times 10^4$, and $1.0 \times 10^4$ genes for φ 5000, 250 and 75 μm irradiations, respectively, out of 22,378 Ensembl protein-coding genes, Supplementary Fig. 2e; random distribution of UMIs and per-gene UMI counts are shown in Supplementary Fig. 2f and g, respectively). We next examined whether the transcriptional profile included spatially specific gene expression. Dimension reduction with uniform manifold approximation and projection (UMAP) revealed that expression profiles from φ 75 μm photo-irradiation areas were clearly separated into three groups according to photo-irradiation site, with the ventromedial and mediomedial genes being closer to each other than the dorsolateral genes (Supplementary Fig. 2h). Differentially expressed genes (DEGs) were detected between dorsolaterally versus mediomedially or ventromedially expressed genes ($n = 198$ or 28, respectively), but not between mediomedially and ventromedially expressed genes (FDR < 0.1, Supplementary Fig. 2i). These results would be considered to reflect their developmental and anatomical origins, in that both mediomedial and ventromedial cells are derived from the apicomedial side of the neural plates before neural tube closure, whereas dorsolateral cells are from the basolateral side. The DEGs were largely separated into the following two clusters (Supplementary Fig. 2j): downregulated in the dorsolateral site (cluster I; $n = 45$) and upregulated in the dorsolateral site, with the latter further branched into two subclusters (clusters II and III; $n = 61$ and 98, respectively). Clusters II and III included several genes known to be expressed in the neural tube, such as *Dcx*, *Hoxb8*, *Zic1*, and *Zic4*. ISH experiments demonstrated that the expression levels of these genes were indeed higher in the dorsal part of the E14.5 neural tube (Supplementary Fig. 2k), consistent with our expression profiling by PIC.

**PIC with RNA-seq under DMD illumination**. To apply PIC for free-form ROIs, a multi-patterned illumination system using digital mirror devices (DMDs) was employed for the PIC photo-irradiation to the CA1 and CA3 areas, and dentate gyrus (DG) of the mouse hippocampus (Fig. 4a). Fresh frozen sections of adult mouse brains were subjected to in situ RT with caged ODNs.

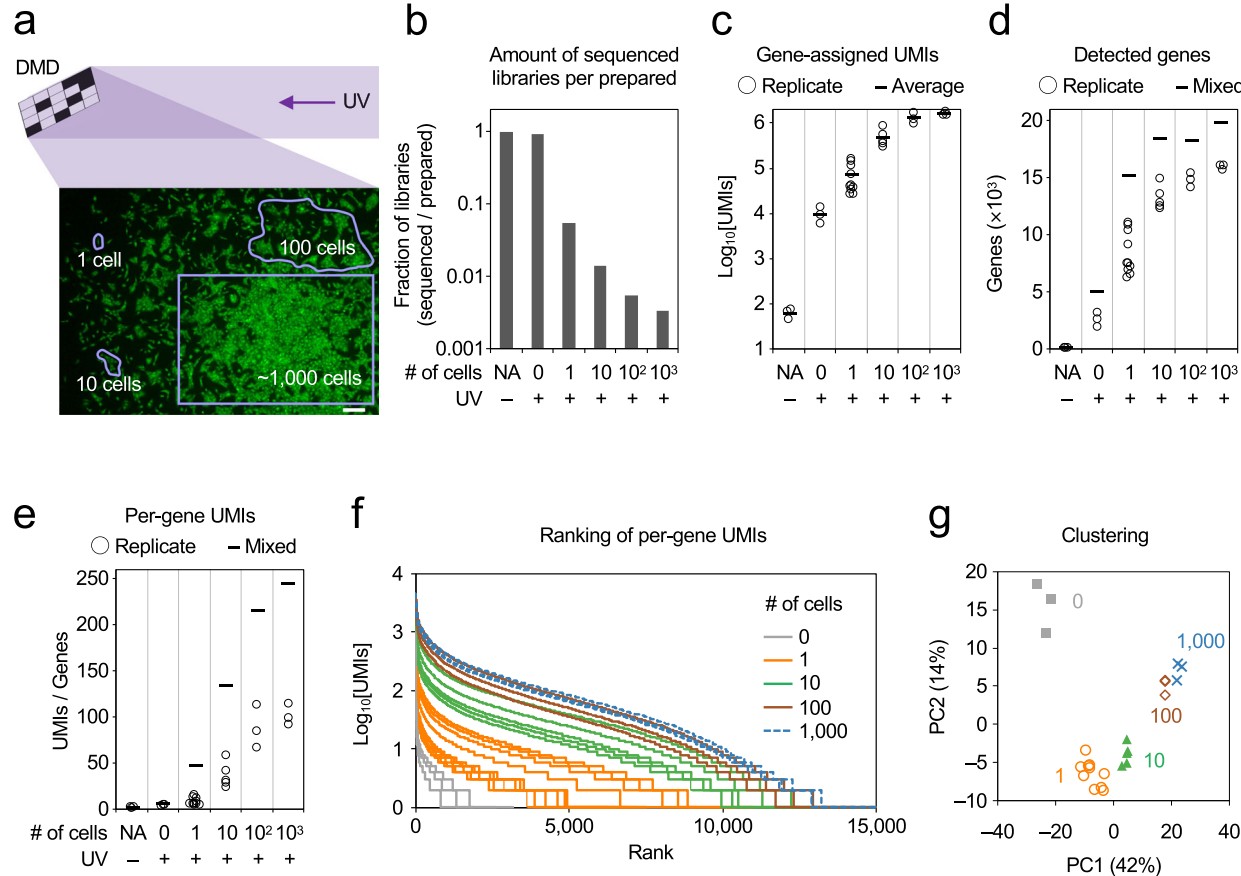

**Fig. 4 Sensitivity of PIC with RNA-seq under DMD illumination. a** Illustration of the experiments for DMD-assisted UV irradiation of 1 ($n=10$), 10 ($n=5$), 100 ($n=3$), or ~1000 ($n=3$) cells. Scale bar, 200 μm. **b** The amounts of the sequenced libraries relative to UV− samples (set as 1) are shown. **c–e** The numbers of gene-assigned UMIs (**c**), detected genes (**d**), and per-gene UMIs (**e**) are shown, with the horizontal lines indicating the average of replicates (**c**) and values when the replicates were mixed (**d**, **e**). **f** Per-gene UMIs and the rankings are shown. **g** Two-dimensional PCA for the expression profiles. Source Data is available as a Source Data file.

After taking nuclear staining images, ROIs were outlined to make "masking sheets", which were loaded on the DMDs so as to irradiate one of the hippocampal regions with 365 nm centred UV light (Fig. 4b). Subsequent RNA-seq detected an average of more than 10,000 genes from one of the hippocampal regions (Fig. 4c), with the numbers of gene-assigned UMIs relative to the total irradiated areas being 50.3, 28.5, and 80.7 UMIs/μm² for CA1, CA3, and DG, respectively (Fig. 4d). Two-dimensional principal component analysis (PCA) indicated that the expression profiles were clearly distinct according to the photo-irradiated hippocampal regions (Fig. 4e). A total of 1070 DEGs were detected (Fig. 4f), including the genes verified by ISH[34] for their specific expression in CA1 (*Wfs1*), CA1 and CA3 (*Ociad2* and *Dkk3*), and DG (*Prox1* and *Pdzd2*).

We evaluated the detection sensitivity of PIC RNA-seq for a small number of cells by DMD irradiation. Five thousand NIH/3T3-GFP cells were seeded onto cover glasses and subjected to in situ RT with caged ODNs containing different barcodes for each replicate. Then, 1, 10, 100, or ~1000 cells (618, 767, and 855 cells) were irradiated with UV by DMDs ($n=10$, 5, 3, and 3, Fig. 4a). In addition, the irradiation was carried out with the ROI area set to zero (0 cells; $n=3$) because a faint level of background illumination was visible outside the ROIs by DMD irradiation. Then, lysate was pooled by the same conditions and the libraries were prepared in accordance with the PIC method described above. Libraries of a sufficient amount were obtained for 1–1000 cells, some of which were sequenced to allow $5 \times 10^6$ reads per

sample (Fig. 4b, Supplemental Fig. 3a). In contrast, the libraries from 0 cells and those without UV irradiation (UV−) were fully sequenced because the amounts in these libraries were extremely small. We found that about half of the reads were assigned to the genes (Supplemental Fig. 3c) and the average UMI counts were $7.0 \times 10^4$, $4.9 \times 10^5$, $1.3 \times 10^6$, and $1.6 \times 10^6$ for 1, 10, 100, or ~1000 cells, respectively (Fig. 4c). In contrast, only $9.6 \times 10^3$ UMIs were detected from the sample with 0 cells, but this would still be an overestimate compared with the other samples due to excessive loading on the flow cell (Fig. 4b). Much lower counts (63 UMIs on average) were detected in UV− samples. The numbers of genes detected in 1, 10, 100, or ~1000 cells were 8349, 13,242, 14,808, and 15,957 in replicate averages, whereas those in UV− and 0 cells were 44 and 2558 genes, respectively (Fig. 4d). The number of detected genes was increased by mixing the replicates; in particular, 15,133 genes were detected by at least one UMI when the gene-assigned UMI count data of 10 samples of single cells were combined. The average per-gene UMI count of single cells was 7.6, and that found by mixing 10 samples was 46.2, which was the same as that of the single replicate of 10 cells (36.3 UMIs/gene on average, Fig. 4e). The minimum number of genes that could be assigned by more than 10 UMIs in a single cell was 427 and the maximum was 3699 (mean 1520 genes, Fig. 4f). In contrast, in the sample with 0 cells, only 59–193 genes (mean 128 genes) were assigned by more than 10 UMIs. Two-dimensional PCA revealed that only the sample with 0 cells was separated from the others (Fig. 4g), suggesting that contamination

by background illumination is trivial and separable, if present at all, from targeted irradiations. These findings indicate that single cells can be covered by multiple UMIs and that combining them increases the coverage proportionally.

**PIC RNA-seq for subcellular and subnuclear microstructures**. Recently, nonmembranous organelles, such as so-called stress granules (SGs) and nuclear speckles (NSs), have been shown to be formed by the aggregation of RNAs and proteins through the physicochemical process termed liquid–liquid phase separation[35–37]. Attempts have been made to survey the transcriptome in the SGs, but the results differed markedly among the different attempts[38–40]. It has been suggested that the conflicting results arose from the different biochemical methods of extracting SGs from other cytoplasmic components, such as cellular lysis, fractionations by density and insolubility, and immunoaffinity, which would result in considerable characteristic changes and loss of the components. Furthermore, transcriptome analysis for NSs has remained challenging due to the difficulty of isolating them from other predominant condensed nuclear components. Hence, we attempted to detect transcripts in SGs and NSs by using PIC with targeted photo-irradiation of those microstructures (Fig. 5a, b). Five thousand HeLa cells were seeded onto coverslips and in situ RT was performed with caged ODNs containing a different barcode for each replicate. The cells were then immunostained with SG and NS markers (G3BP1 and SC35, respectively) and the resulting immunofluorescent images were binarised at an appropriate threshold to make "masking sheets" to irradiate them (designated as SG-posi and NS-posi, respectively). As the controls for SG-posi, we created masking sheets to irradiate the cytoplasm except for SGs and nuclei (SG-nega) or the cytoplasm including both SG-posi and SG-nega regions (Both). In addition, masking sheets as controls for NS-posi were made so as to irradiate nuclei except for NSs (NS-nega) or the nuclei (Both). These masking sheets were loaded in the DMD system, and were capable of precise irradiations to the SGs and NSs, as well as to other control regions (Fig. 5a, b). On the basis of a preliminary simulation, photoirradiation of 200 SGs and NSs had been estimated to obtain a sufficient S/N ratio (Supplementary Fig. 4), but an extended number of cells (~100 cells encompassing several hundred SGs and NSs) were actually irradiated with each biological replicate. Lysates were pooled from five replicates and the library was prepared in accordance with the PIC method as described above. Sequencing was performed so as to obtain $5 \times 10^6$ reads per sample with a portion of the library (Fig. 5c, Supplementary Fig. 3b). As a result, 44% and 46% reads were assigned to genes for SG-posi and NS-posi samples, respectively, and were similar to the other samples (Supplementary Fig. 3d, e). The average gene-assigned UMI counts were $2.4 \times 10^5$ and $2.2 \times 10^5$ for SG-posi and NS-posi samples, respectively, and the average UMI count for other irradiated regions was also in a similar range, but that for UV– was limited to about 1% of the level (Fig. 5d). The numbers of genes detected in SG-posi and NS-posi were 10241 and 10761 on average, and the number of genes in the other irradiated areas was similar (Fig. 5e). The number of gene-assigned UMIs relative to the total irradiated areas was calculated to be 143 UMIs/$\mu m^2$ for SG-posi and 318 UMIs/$\mu m^2$ for NS-posi (Fig. 5f), indicating that more than 100 transcripts were detectable within a $1 \times 1 \, \mu m$ region by PIC RNA-seq. Two-dimensional reduction of the expression profiles using UMAP suggested that SG-posi and NS-posi were more distinct than the others (Fig. 5g). GSEA analysis (MSigDB C2 collection) showed that genes abundant in SG-posi and NS-posi samples were enriched in many more gene sets relative to the associated controls, including transcripts encoding ribosomal proteins and others involved in translational mechanisms (Supplementary Fig. 5a, b). In contrast, few gene sets were significantly enriched in transcripts in SG-nega and NS-nega samples (FDR = 0.1). DEG analysis revealed a total of 1655 genes in SG-posi vs. SG-nega and 1990 genes in NS-posi vs. NS-nega samples (FDR = 0.1, Fig. 5h). A simulation test with random reduction of replicates showed that more than 300 DEGs were still detected if reduced to three replicates (Supplementary Fig. 5c), indicating that our experimental designs were reasonable to obtain a sufficient number of DEGs. Notably, *MALAT1*, the most abundant constituent RNA of NS, was detected in NS-posi-specific DEGs (Fig. 5h), among which *MALAT1* and *AKR1C2* transcripts were shown to be localized in, or at the periphery of, NSs by fluorescent ISH analysis (Fig. 5i).

## Discussion

We demonstrated that PIC with RNA-seq for tissue sections could detect the expression of approximately 10,000 genes from only a few dozen cells from the mouse neural tube. Furthermore, ~200 genes were identified to be expressed with spatial specificity according to the photo-irradiation sites. Furthermore, more than 10000 genes were detected from 100 cells by UV irradiation to intracellular and nuclear structures, thereby highlighting hundreds of transcripts specifically localised in SGs and NSs. This technology thus enables gene expression profiles to be obtained from small-scale ROIs. One of the most critical issues of PIC development is to suppress the background amplified from nonirradiated regions. Such background is likely to originate from the read-through of DNA polymerase at the NPOM-dT sites of the template strand[28]. Repetitive insertion of caged nucleotides was found to be effective at decreasing the background to as little as less than 0.1% relative to that in photo-irradiated samples. The use of the optimised caged ODNs reduced the background to under the detection limit of qPCR for in situ RT samples from cell cultures and tissue sections. The background of RNA-seq reads was 0.4% relative to that of φ 75 μm irradiated samples (= an area of 4400 μm²), although the area of nonirradiated cells in E14.5 sections was several thousand-fold larger (areas of 10–20 mm²). These findings indicate that the amount of background is low, at least for specimens several millimetres in size. LCM has been used to analyse spatially localised cells by the physical isolation of ROIs with a high-power laser[41], and LCM-seq is one of the most sensitive applications capable of detecting gene expression from single cells[21]. In general, the ROI size and sharpness of the cut edge of LCM depend on the laser power and width as well as the stiffness of the specimen. PIC is conceptually different from LCM in the sense that ROIs are photochemically isolated. The theoretical spatial resolution of photo-irradiation is up to the diffraction limit as well as that of Nano-String GeoMx Digital Spatial Profiler[22]; therefore, these technologies can be applied to submicron-sized ROIs, with the sharpness of the edge being extremely fine. Circular areas can be irradiated using a conventional fluorescence microscope. Furthermore, regions with complicated shapes can be irradiated using pattern illumination systems, such as DMD and galvo-based scanning systems[42,43]. PIC is thus suitable to analyse cells in complicated fine structures, such as renal glomeruli, which are composed of mesangial cells, podocytes, endothelial cells, Bowman's capsule, and proximal tubules[44]. PIC can also be applied to analyse cells in a histologically scattered pattern, such as Sertoli cells in the testes or lymphocytes in inflamed tissues[45]. Transcripts localised in subcellular and subnuclear components are also favourable targets, including the organelles, subcellular microdomains of neurons, and the subnuclear chromosome territories. Therefore, PIC can dissect out detailed characteristics of cells with fine spatial resolution.

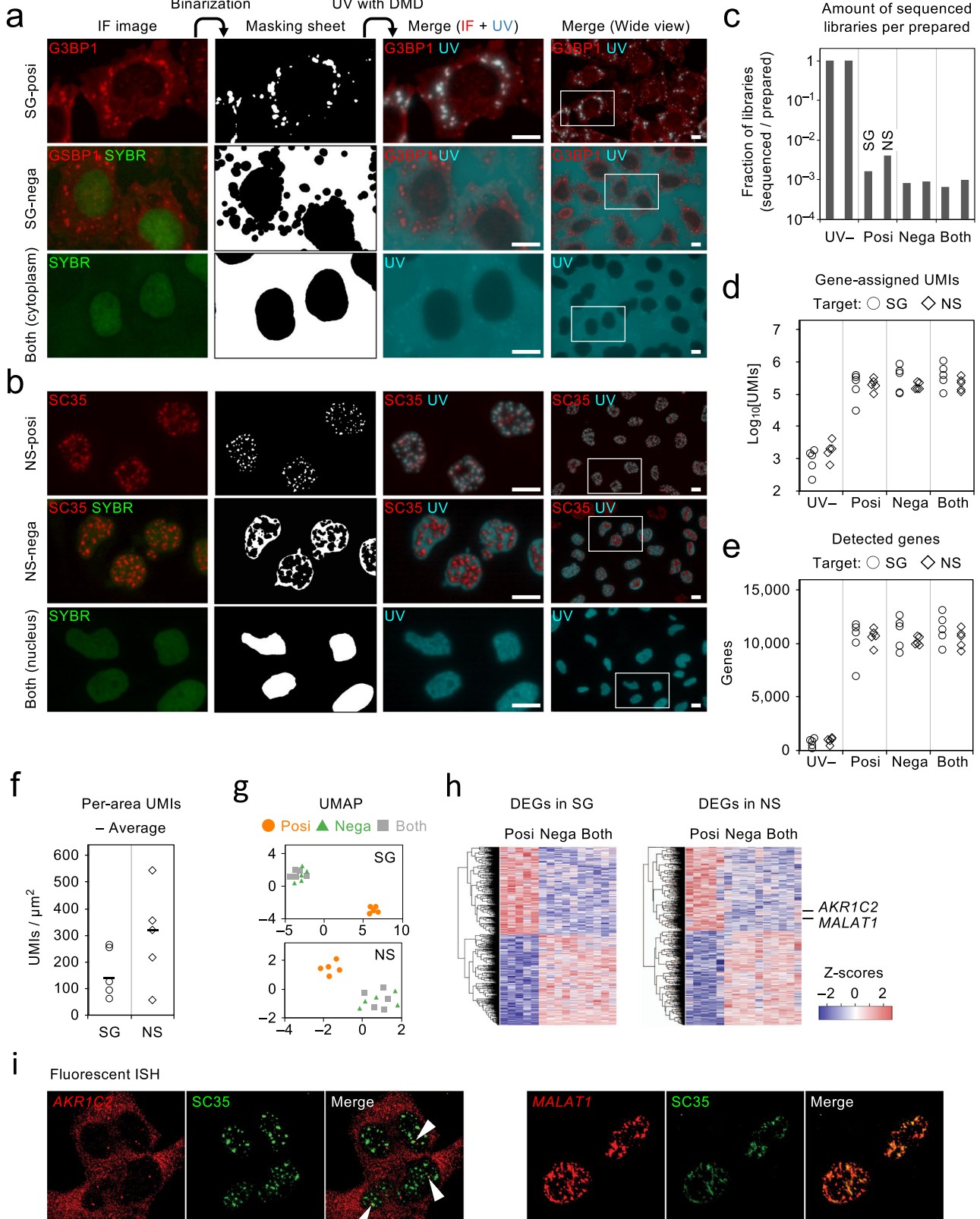

**Fig. 5 PIC RNA-seq for subcellular and subnuclear microstructures. a**, **b** Images of green nuclei, red immunofluorescence (IF) for SGs (**a**) and NSs (**b**), resulting in masking sheets, and merged images of IF and DMD-assisted UV irradiation are shown. A representative image was shown out of five replicates. **c** The amounts of the sequenced libraries relative to UV– samples (set as 1) are shown. **d**–**f** The numbers of gene-assigned UMIs (**d**), detected genes (**e**), and per-area UMIs (**f**) are shown, with the horizontal lines indicating the average of replicates (**f**). **g** Two-dimensional reduction by UMAP analysis for the expression profiles. **h** DEGs are clustered and shown in heat maps. **i** Fluorescent ISH for *AKR1C2* and *MALAT1* transcripts. A representative image was shown out of three replicates. Scale bars, 10 μm. Source Data is available as a Source Data file.

To understand cellular characteristics and the spatial interaction of cells at the tissue level, it is necessary to characterise a wide variety of cells by deep expression profiling in combination with high-resolution spatial information. In this context, PIC can profile over 10000 genes from several dozen cells from limited areas. Single-cell transcriptome analysis can also be performed by pooling multiple barcodes, by which throughput for library preparation was markedly elevated. The spatial resolution of PIC is up to subcellular and subnuclear levels and the sequencing depth is high, but the throughput to analyse multiple regions is low. LCM has an advantage in terms of multiplexity where multiple ROIs clipped from an identical section can be separated into some distinct test tubes. The NanoString platform can detect a panel of genes with higher spatial resolution than LCM, together with an advantage in throughput that multiple ROIs can be separated from an identical section by sequentially repeating photo-irradiation and aspiration. A wide range of cell types can be analysed by multiplexed profiling with spatial transcriptomics, Slide-seq, and HDST technologies[16–18]. The resulting cell types of interest may be further analysed by PIC to obtain expression profiles at much higher depth. Transcriptome in vivo analysis (TIVA) is an alternative to PIC if targeting living cells[46]. PIC can also detect spatially heterogeneous cells in tissues that appear to be homogeneous, such as different spatial domains of the neural tube, as demonstrated in this study. The subcellular localisation of transcripts may be analysed thoroughly by seqFISH +, MER-FISH, FISSEQ, and ExSeq at single-molecule resolution[14,15,19,20]. PIC can also be used for subsequent higher-depth sequencing for the targeted subcellular domains. Therefore, PIC is a powerful tool when combined with other spatial transcriptome analyses for understanding multicellular to subcellular systems. For the further development of spatial transcriptomics, it will be useful in the future to benchmark and compare various techniques including PIC at the single-cell level. One of the exclusive advantages of PIC is that it can be performed with standard laboratory equipment. In addition to standard CEL-seq2, the only essential material is the caged ODNs, which can be synthesised by general oligo-vendors, and NPOM-caged dT is commercially available. Once synthesised, the caged ODNs are used for hundreds of in situ RT reactions, with the per-reaction cost being only a few dollars. PIC is thus generally easy to introduce and should be the first choice for deep transcriptome analysis in small ROIs.

In multicellular systems, spatially specific gene expression is orchestrated by chromatin conformation and epigenetic modifications. These regulatory systems have been studied at the genome-wide level to investigate open chromatin, protein-genome binding, and genome methylation, but it remains challenging to perform such analyses on cells located in a limited area. Given that some technologies, such as ATAC-seq, ChIL-seq, and PBAT (post-bisulfite adapter tagging), commonly use ODNs as a starting material[47–50], caged ODNs provide the advantage of being able to amplify sequence libraries only from photo-irradiated ROIs. Thus, PIC will be able to determine epigenetic landscapes as well as expression profiles in an ROI-specific manner. Multiple cell types that can be juxtaposed or located at a distance from one another can mutually interact. Simultaneous analysis of such interacting cell types will be enabled by the combined use of ODNs caged with NPOM and other caging groups having distinct wavelength-selectivity[51–53]. After the in situ RT with such mixed ODNs, multi-colour irradiation and barcode sequencing will separate the sample information. Multi-colour PIC will also be useful to compare transcripts showing distinct intracellular localisation. Further improvement of the sensitivity and protocols of PIC would be necessary for such innovations.

## Methods

**Cells**. NIH/3T3-GFP and T-47D cells were cultured at 37 °C in a humidified 5% $CO_2$ atmosphere, in high-glucose DMEM (Gibco; 11965092) supplemented with 10% calf serum, 1% penicillin/streptomycin (Nacalai Tesque), and 1% L-glutamine (Nacalai Tesque). HeLa cells were cultured in the same conditions with an EMEM-based culture medium (KAC Co., Ltd.; DSGM105). To induce the expression of GFP, 1 μg/μl doxycycline (Sigma-Aldrich) was added to NIH/3T3-GFP cells. For sample preparation, cells were dissociated by trypsinisation and then inoculated onto gelatinised coverslips either directly or after the formation of cell aggregates by the hanging drop culture method (1000 cells per 20 μl medium drop for 2–3 days). Total RNA was isolated using an RNeasy Micro Kit (Qiagen). To form SGs, HeLa cells were incubated for 30 min with 75 mM $NaAsO_2$ (Sigma) in the culture medium before fixation.

**Mice**. The study was approved by the Animal Care and Use Committee of Kyushu University. ICR mice were purchased from Charles River Laboratories Japan. E14.5 embryos and 8-week-old male brains were dissected, then immediately embedded in O.C.T compound (Sakura) and immersed in isopentane/dry ice for flash-freezing. Cryosections at a thickness of 10 μm were mounted on MAS-coated glass slides (Matsunami) and air-dried.

**Caged ODNs**. NPOM-caged dT-CE phosphoramidite was purchased from Glen Research (10-1534-95) and used to synthesise caged ODNs with OPC-grade purification by Nihon Gene Research Laboratory. The synthesised caged ODNs were shielded from the light during transport. After receipt, a solution of caged ODNs was immediately aliquoted into single-use volumes and freeze-stored in a light-shielded box. The following steps of the in situ RT and immunostaining were performed in light-shielded humidified chambers, and UV irradiation was performed in a dark room, but the other steps were conducted under standard room light.

### PIC protocol

*Fixation and permeabilisation*. Cells on coverslips or tissue sections were washed twice with PBS-diethylpyrocarbonate (DEPC) and fixed with 3.7% formaldehyde solution in PBS-DEPC for 10 min at room temperature. Specimens were permeabilised with 5% TritonX-100 in PBS-DEPC for 3 min and then with 0.1 N HCl for 5 min, followed by neutralisation with 1 M Tris-HCl, pH 8.0, for 10 min at room temperature.

*In situ* RT. Permeabilised specimens were incubated in PBS-DEPC for 5 min at 65 °C and quickly cooled in ice-cold PBS-DEPC. Primer mix [0.5 μl of 500 ng/μl caged ODNs with a distinct barcode (Supplementary Table), 0.5 μl of 10 mM dNTPs (NEB) and 5 μl of $H_2O$] was also heated to 65 °C and then quickly cooled to 4 °C, before combining with first-strand mix (2 μl of 5 × First-Strand Buffer, 1 μl of 0.1 M DTT, 0.5 μl of 40 U/μl RNase Out, and 0.5 μl of 200 U/μl Superscript II Reverse Transcriptase, all from Invitrogen). The RT reaction mix was applied to the specimens, which were then coverslipped, incubated at 42 °C for 60 min, and heat-inactivated in 70 °C PBS for 10 min.

*Immunostaining*. Specimens were blocked with blocking solution [50% Blocking OneP (Nacalai Tesque) in TBST (Tris-buffered saline with 0.1% Tween)] for 10 min at room temperature, incubated with an anti-SOX2 rabbit monoclonal antibody (Cell Signalling #23064; 1:1000) overnight at 4 °C and then incubated with an Alexa488-conjugated anti-rabbit IgG antibody (Invitrogen; 1:1000; for 1 h at room temperature). After nuclear staining with TOPRO3 (1:1000 in TBST), specimens were mounted and coverslipped with 90% glycerol containing 1/1000 TOPRO3 and 223 mM 1,4-diazabicyclo[2.2.2]octane (DABCO). HeLa cells were blocked under similar conditions and incubated with rabbit anti-G3BP1 antibody (Novus; NBP1-18922; 1:1000) or mouse anti-SC35 antibody (Abcam; ab11826; 1:1000) for 1 h at room temperature. To visualise SGs (G3BP1), cells were incubated for 1 h at room temperature with both donkey anti-rabbit IgG Alexa555plus (Invitrogen; 1:1000; to visualise SGs) and goat anti-rabbit IgG Alexa405 (Invitrogen; 1:250; to visualise UV-irradiated SGs), together with 1/1000 SYBR Green I (Invitrogen; 1:1,500,000; to visualise nuclei). To visualise NSs (SC35), cells were incubated for 1 h at room temperature with both donkey anti-mouse IgG Alexa555plus (Invitrogen; 1:2000; to visualise NSs) and DAPI (Nacalai; 1:2000; to visualise UV-irradiated nuclei), together with 1/1000 SYBR Green I (Invitrogen; 1:1,500,000; to visualise nuclei).

*Photo-irradiation*. Photo-irradiation of cell cultures and tissue sections for uncaging was performed under a Leica DM5000 B fluorescence microscope illuminated with an EL6000 100 W Hg lamp (100% power) through a Leica HCX objective lens [2.5×/0.07 Plan (506304), 5×/0.15 PL S-APO (506288), 10×/0.30 PL S-APO (506289), 20×/0.50 PL S-APO (506290), 40×/0.75 PL S-APO (506291) or 100×/1.40-0.70 OIL PL APO (506210)] and a Leica A filter cube (11531873) at a wavelength of 340–380 nm for 15 min, unless otherwise indicated. Fluorescence field diaphragms were set at levels 1 and 2 for photo-irradiation of φ16 and 75 μm areas, respectively, with a × 100 lens. Photo-irradiation of solutions in test tubes was performed under a Nikon C1 fluorescence microscope illuminated with a C-HGFI 100 W Hg lamp (100% power) through a 20×/0.45 S Plan Fluor (MRH48230) objective lens and a Semrock DAPI-5060C-NTE filter cube at a

wavelength of 352–402 nm for 15 min. DMD-assisted photo-irradiation was performed with a multi-pattern LED illumination system (Opto-line; LEOPARD2) equipped with a DMD (Mightex; Polygon 1000), dichroic mirror (Semrock; 442 nm; Di02-R442-25×36) and LED light source (Prizmatix; UHP-F-365 LED; 3 W; 100% power) under a Leica DM6B fluorescent microscope through a Leica objective lens using HCX PL FLUOTAR 10×/0.32 (506521; for UV irradiation to the hippocampus), HCX PL FLUOTAR 20×/0.55 (506519; for UV irradiation to 1 to ~1000 cells), or HC PL APO 63×/1.40-0.60 OIL (506349; for UV irradiation to SGs and NSs) for 15 min. Masking sheets loaded on the DMDs were made by ImageJ Fiji (version 2.1.0).

*Cell lysis.* Lysis solution (0.1% Tween and 400 μg/ml Proteinase K in PBS) was loaded onto specimens and incubated for 30 min at 55 °C. Lysates of the replicates with distinct barcodes were combined into a single tube. cDNA:mRNA hybrids were purified with a Qiagen MinElute PCR Purification kit and eluted with $H_2O$.

*Second-strand DNA synthesis.* The eluted cDNA:mRNA hybrids (15 μl) were combined with a second-strand mix [2 μl of 5× First-Strand Buffer (Invitrogen), 2.31 μl of Second Strand Buffer (Invitrogen), 0.23 μl of 10 mM dNTPs (NEB), 0.08 μl of 10 U/μl E. coli DNA ligase (Invitrogen), 0.3 μl of 10 U/μl E. coli DNA polymerase (Invitrogen) and 0.08 μl of 2 U/μl E. coli RNaseH (Invitrogen)] and incubated for 2 h at 16 °C. The double-stranded cDNAs were purified with AMpure XP beads (Beckman Coulter) and eluted with $H_2O$.

*IVT.* The eluted double-stranded cDNAs (6.4 μl) were combined with IVT mix [1.6 μl each of A/G/C/UTP solution, 1.6 μl of 10× T7 reaction buffer, and 1.6 μl of T7 enzyme from the MEGAscript T7 Transcription Kit (Invitrogen)] and incubated for 17 h at 37 °C. One microliter of 2 U/μl TURBO DNase (Invitrogen) was added and incubated for 15 min at 37 °C. Six microliters of ExoSAP-IT PCR Product Cleanup Reagent (Invitrogen) were added and incubated for 15 min at 37 °C, after which 5.5 μl of fragmentation buffer (200 mM Tris-acetate pH 8.1, 500 mM KOAc, and 150 mM MgOAc) was added and incubated for 3 min at 94°C. After adding 2.75 μl of 0.5 M EDTA pH 8.0, the aRNAs were purified using RNAClean XP beads (Beckman Coulter) and eluted with $H_2O$.

*Library preparation and sequencing.* The eluted aRNAs (5 μl) was combined with RT primer mix [1 μl of 250 ng/μl randomhexRT primer (Supplementary Table) and 0.5 μl of 10 mM dNTPs (NEB)], incubated for 5 min at 65 °C and then quickly cooled to 4 °C. RT reaction mix was then added [2 μl of 5× First-Strand Buffer, 1 μl of 0.1 M DTT, 0.5 μl of 40 U/μl RNase Out and 0.5 μl of 200 U/μl Superscript II Reverse Transcriptase (all from Invitrogen)], incubated for 10 min at 25 °C and further incubated for 1 h at 42 °C. The RT products (9 μl) were combined with PCR mix [1.8 μl each of 10 μM RNA PCR primers 1, 2, and 22.5 μl of Phusion High-Fidelity PCR Master Mix and 9.9 μl of water], and amplified by PCR (98 °C for 30 s, followed by 11 cycles of 98 °C for 10 s, 60 °C for 30 s and 72 °C for 30 s, with a final extension at 72 °C for 10 min). Fragments of 250–500 bp were then purified and size-selected with AMpure XP beads. The quality of the resulting cDNA library was assessed using high-sensitivity DNA chips on a Bioanalyzer 2100 (Agilent Technologies) and quantified using a Library Quantification Kit (Clontech), before sequencing on the Illumina HiSeq 1500 platform.

*Data analysis.* The barcodes and UMIs in the reads were extracted using UMI-tools (version 1.0.0) with the following command: umi_tools extract -I r1.fastq --read2-in=r2.fastq --bc-pattern=NNNNNNCCCCCC --read2-stdout. The reads were mapped by the aligning software HISAT2 (version 2.1.0) to the reference genome (GRCh38 and GRCm38). Read counts per gene were determined with featureCounts (version 1.6.4) and UMI-tools with the following command: umi_tools count --method=directional --per-gene --per-cell --gene-tag=XT. The duplicates in the unique molecular identifiers were discarded. Using the resulting counts, differentially expressed genes (FDR = 0.1) were extracted by the R library DESeq2 (version 1.20.0), which was also used to transform count data into regularised log data before clustering (heatmap3, version 1.1.6) and dimension reduction (UMAP, version 0.2.2.0).

**UV-dependent primer extension.** Here, 50 μM each of caged ODNs and tetramethylrhodamine (TAMRA)-labelled primer (Supplementary Table) were mixed and incubated for 1 min at 95 °C and then gradually cooled to 25 °C for 45 min using a thermal cycler. Half of the annealed oligonucleotides were photo-irradiated for 15 min with 352–402 nm wavelength light as mentioned above. The annealed ODNs with or without photo-irradiation were subjected to the chain extension reaction [final concentration of 1.42 μM annealed ODNs, 1× Second buffer, 0.5 mM dNTPs and 10 U/μl E. coli DNA polymerase I (Invitrogen)] for 2 h at 16 °C, followed by heat denaturation for 1 min at 98 °C. Reaction aliquots of 5 μl were electrophoresed on denaturing urea polyacrylamide gels (15%) and TAMRA fluorescence was detected by a UV transilluminator (TOYOBO, FAS-201).

**qPCR.** Quantification of several genes in sequence libraries was performed by real-time PCR using NEBnext PCR Master Mix (NEB) supplemented with SYBR Green (Applied Biosystems). Other qPCR experiments were performed using TaqMan Gene Expression Master Mix (Applied Biosystems). Primers are shown in the Supplementary Table.

**ISH.** Digoxigenin-labelled riboprobes were synthesised with DIG RNA Labelling Mix (ROCHE), Thermo T7 RNA Polymerase (ToYoBo), and gene templates cloned using T7 promoter-containing PCR primer sets (Supplementary Table) and mouse embryonic brain cDNA pools. Fresh-frozen sections of E14.5 mouse embryos were washed twice with PBS-DEPC and fixed with 4% paraformaldehyde for 12 h at 4 °C. The sections were then serially treated with 6% $H_2O_2$ for 20 min, 10 μg/ml proteinase K solution for 5 min, and post-fix solution (4% paraformaldehyde, 0.2% glutaraldehyde, and 0.1% Tween in PBS-DEPC) for 20 min at room temperature. After incubation with prehybridisation mix (50% formaldehyde, 5× SSC pH 5, 150 μg/ml yeast RNA, 150 μg/ml heparin, 1% SDS, and 0.1% Tween) for 30 min at 65 °C, hybridisation was performed overnight with the same solution containing a digoxigenin-labelled riboprobe. Sections were then washed several times with SSC of increasing stringency and subsequently incubated with alkaline phosphatase-conjugated anti-digoxigenin antibodies (Roche). Nitro blue tetrazolium and 5-bromo-4-chloro-3-indolyl-phosphate (Roche) were used for the colorimetric detection of alkaline phosphatase activity, followed by nuclear staining with 4′,6-diamidino-2-phenylindole (DAPI; Invitrogen). Both bright-field and nuclear staining images were separately taken using a Keyence BZ-X700 All-in-One microscope.

**Reporting summary.** Further information on research design is available in the Nature Research Reporting Summary linked to this article.

## Data availability
Deep sequencing data in this study have been deposited in the Gene Expression Omnibus (GEO) under accession code GSE143413. Source data are provided with this paper.

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

## Acknowledgements

We are grateful to the staff of Ohkawa/Meno Laboratory in Kyushu University and Yumi Sugahara and Prof. Shu Narumiya in Kyoto university for technical support and discussion, and also thank the Advanced Computational Scientific Programme of the Research Institute for Information Technology, Kyushu University. We additionally thank Jeremy Allen, PhD, from Edanz (https://jp.edanz.com/ac) for editing a draft of this manuscript. This work was supported by MEXT/JSPS KAKENHI (JP19K22398 to M.H.; JP19H03424 to S.O.; JP25116010, JP17H03608, JP17K19356, JP18H04802 and JP18H05527 to Y.O.; JP16H01219, JP15K18457 and JP18K19432 to A.H.; JP16K18479, JP16H01577, JP16H01550 and JP18H04904 to K.M.), JST ACT-X (JPMJAX201D to M. H.), JST PRESTO (JPMJPR1942 to S.O.) and JST CREST (JPMJCR16G1 to Y.O.).

## Author contributions

S. O. and Y. O. conceived PIC and supervised all aspects of the work. M. H., R. K., and S. O. performed the experiments. A. H. and C. M. assisted with the experiments. S. O., K. M., and K. T. analysed the sequencing data. M. H., S. O., and Y. O. wrote the paper. All authors read and approved the final paper.

## Competing interests

S.O. and Y.O. are involved in a pending patent related to PIC technology (application number, 2019-094216; patent office, Japan Patent Office; current status, patent pending). All other authors declare no competing interests.
