## [Peer Review File · Nature Communications]

Reviewers' Comments:

Reviewer #1:

Remarks to the Author:

The authors response to the prior reviews is appreciated, and while some issues have been dealt with appropriately there are still multiple concerns with regard to the technology as detailed in the attached comments.

The number of genes assigned by 10 or more UMIs ranged from 427 to 3699 with an average of 1520 genes. This range of detection is a nearly 10-fold difference and suggests difficulty in reproducibility of the technique. Further if mixing these 10 replicates one gets 15,133 different genes then this suggests that the technique is for some reason missing many high abundance RNAs (10 or more UMIs) for any particular cell.

In discussing applications of PIC the authors comment on use of PIC using random primer sequence to be able to assess mRNA splice forms, but this will likely be quite difficult at best as the primers would certainly pick up mostly ribosomal RNAs decreasing the number of reads for any particular mRNA. This is the trick for CEL-seq (concentrating on poly-A mRNA) which is one of the reasons it works.

Even though the MM and VM domains are from the same development area of the neural plate, the fact that MM and VM domains could not be distinguished by DEGs using PIC is concerning. This suggests that the PIC technology is not robust or reproducible enough to show this distinction.

While the citations have improved, it is important to show the foundational papers that lead to any new datasets. There are many other papers showing cDNA synthesis on fixed tissue that have not been cited, including FISSEQ (first step is in situ cDNA synthesis) and Crino et al, etc.

The authors response to the worry about small RNAs acting as primers is welcome, but unfortunately not sufficient. It is unfortunately still a problem as T7 RNA polymerase can cause nonspecific RNA amplification independent of T7 promoter function due to binding to nonspecific 3' overhangs on double stranded DNA (generated during the molecular biology steps) and acting as sites for nonpromoter initiated RNA synthesis. While it is true these would not have the UMI specifically introduced by cDNA synthesis, dependent upon the parameters for calling a sequencing read a real read (% error tolerated, etc.), a region could mistakenly be called a UMI. This is potentially a serious issue that would complicate the interpretation of these data.

In activation of the oligonucleotide, Irradiation at 100 W at 100% power for 15 min. will clearly cause unwanted damage to RNA, DNA and tissue (there are several papers that highlight this including Klak et al., "Irradiation with 365nm and 405nm wavelength shows differences in DNA damage of swine pancreatic islets"). It will produce free radical formation (even in fixed tissue samples) causing crosslinking, oxidation, etc. This issue would be lessened if the uncaging time was reduced. Certainly for most such photoactivatable compounds the efficiency of uncaging should be milliseconds. The use of a 15min time window with 100% power is quite concerning.

Contrary to the authors response, loss of RNA upon heating may still be an issue as getting more amplified RNA doesn't necessarily mean less loss of RNA but as mentioned previously by the reviewer and the author, perhaps more RNA accessibility which would allow longer cDNA synthesis and longer amplified RNAs giving a higher yield, not necessarily showing little loss of cellular RNAs.

The qPCR for Gapdh does not address the reproducibility of the overall sample but rather a single

RNA. In the described control experiment, It is difficult to envision how RNA from a fixed tissue section will yield more signal (Gapdh PCR signal was "comparable to, or slightly higher") than cell isolated RNA assessed by traditional CEL-seq2.

The additional data showing 10,000 genes in SG and nuclear speckles by PIC is also troubling. As the site of precursor RNA processing, nuclear speckles will contain 10-100x less RNA than the cytoplasm where RNA stability allows RNA abundances to be high (hnRNA is always significantly less than cytoplasmic RNA). Further, technically it should be much harder to detect RNAs in nuclear speckles (given their lower amounts) and hence there should technically (not just biologically) be fewer genes identified then from cytoplasmic RNA. Why there would be a DEG of 1990 genes for NS-positive vs NS-negative samples is also curious as they should be very similar in composition as the NS is the processing site for cytoplasmic RNAs. But as highlighted earlier the detectability will be less as there should be less RNA detected in NS than the cytoplasm.

Finally, it is a minor point but the discussion of differences in rebuttal to reviewer 1, between PIC and LCM are misleading as they both rely upon laser characterization of the area to be transcriptionally assessed and as long as the area can be visualized there should be little loss due to complicated morphologies.

Reviewer #2:

Remarks to the Author:

Honda and colleagues have developed a method of great utility for spatial gene expression profiling using photo-caged oligos (caged ODNs) with single cell resolution.

The authors have optimized a number of parameters to improve the method and now even show utility down to single cells.

This manuscript is much improved compared to the previous version and all my concerns have been addressed. Furthermore the possibility to conduct clipping of subcellular compartments is very attractive and I believe this method will be very useful across research fields.

Reviewer #3:

Remarks to the Author:

The authors have answered the comments from reviewer 3 in an appropriate manner. The addition of single cell, 10 cell etc data to the manuscript is a great improvement. The technology displays a high sensitivity, capturing a lot of UMI and genes, not seen in similar methods, as the authors point out.

The title has been updated to include "High-depth" which does not sound grammatically sound to me. Perhaps "High sensitivity" is better, but a native English speaker would know best.

Reviewer #1:

Honda and colleagues have developed a method of great utility for spatial gene expression profiling using photo-caged oligos (caged ODNs) with single cell resolution.

The authors have optimized a number of parameters to improve the method and now even show utility down to single cells.

This manuscript is much improved compared to the previous version and all my concerns have been addressed. Furthermore the possibility to conduct clipping of subcellular compartments is very attractive and I believe this method will be very useful across research fields.

We thank the reviewer for providing us with critical comments and for accepting our manuscript.

Reviewer #2

The authors response to the prior reviews is appreciated, and while some issues have been dealt with appropriately there are still multiple concerns with regard to the technology as detailed in the attached comments.

The number of genes assigned by 10 or more UMIs ranged from 427 to 3699 with an average of 1520 genes. This range of detection is a nearly 10-fold difference and suggests difficulty in reproducibility of the technique. Further if mixing these 10 replicates one gets 15,133 different genes then this suggests that the technique is for some reason missing many high abundance RNAs (10 or more UMIs) for any particular cell.

It is a common finding that the numbers of detected genes can vary by up to 10-fold among low-input transcriptome samples, as has been reported in benchmark papers that compared the performances of multiple single-cell RNA-seq technologies, including CEL-seq2 (Ding, et al., 2020. *Nat Biotechnol*; Mereu, et al., 2020. *Nat Biotechnol*). Therefore, it is reasonable to expect that photo-isolation chemistry (PIC)-coupled RNA-seq, which is based mainly on CEL-seq2, will have a similar range of variance as the other technologies that were developed to amplify small numbers of transcripts.

The concern that many high abundance RNAs may be missed (last sentence, underscored) seems to be caused by a misunderstanding of a sentence in our previous manuscript. We wrote “*15,133 genes were detected when 10 samples of single cells were combined*” to mean that 15,133 genes were assigned by one or more unique molecular identifiers (UMIs) by combining all UMI count data from 10 replicates of single cells. For clarity, we have revised the sentence in the RESULT section as: “*15,133 genes were detected by at least one UMI when the gene-assigned UMI count data of 10 samples of single cells were combined.*”

In discussing applications of PIC the authors comment on use of PIC using random primer sequence to be able to assess mRNA splice forms, but this will likely be quite difficult at best as the primers would certainly pick up mostly ribosomal RNAs decreasing the number of reads for any particular mRNA. This is the trick for CEL-seq (concentrating on poly-A mRNA) which is one of the reasons it works.

This comment seems to be because of a misunderstanding of the following sentences in the DISCUSSION section of our previous manuscript:

“RT primers capturing non-poly-A RNAs are useful to detect a wide variety of noncoding RNAs and splicing variants such as those developed in RamDA-seq technology. If caged, such RT primers will be applied for unbiased deep sequencing in photo-irradiated regions.”

Random displacement amplification sequencing (RamDA-seq) (Hayashi, et al. 2018. *Nat Commun*) was developed to detect non-ribosomal RNAs using “semi-random” primers that are devoid of ribosomal RNA sequences. Therefore, RamDA-seq can capture any non-ribosomal RNAs, including mRNAs from 5' to 3' regions, introns, and noncoding RNAs without poly-A tails. This technology is suitable for low-input transcriptomics and the semi-random primers are commercially available. Therefore, photo-caging of the semi-random primers is considered useful for region of interest (ROI)-specific detection of all types of RNAs, except ribosomal RNAs. To avoid any further confusion, we removed these sentences from the DISCUSSION because of the lack of data to support the idea.

Even though the MM and VM domains are from the same development area of the neural plate, the fact that MM and VM domains could not be distinguished by DEGs using PIC is concerning. This suggests that the PIC technology is not robust or reproducible enough to show this distinction.

In our previous response letter, we noted that the medio-medial (MM) and ventro-medial (VM) cells originated from the apico-medial side of the neural plate, whereas dorso-lateral (DL) cells are from the baso-lateral side. Furthermore, in E14.5 neural tubes, MM and VM cells both show common characteristics as neural progenitor cells, whereas DL cells differentiate into neuronal lineage. Thus, the experiment shown in Fig. 3 was designed to compare the similarity of MM and VM cells and to show their differences to DL cells. As expected, differentially expressed genes (DEGs) were detected between DL and MM/VM cells, but not between MM and VM cells. Nevertheless, replicates from the MM and VM cells were clustered into distinct subtypes by UMAP, indicating that small differences among multiple genes were detected in a “reproducible” manner. Therefore, PIC RNA-seq is considered to be “robust” against the noise that accompanies low-input transcriptome analyses. In the revised manuscript, we described only the observed results and avoided using the words “reproducible” or “robust” as was done in the previous manuscript, because these two words are not quantitative and could be viewed as overstating the findings.

While the citations have improved, it is important to show the foundational papers that lead to any new datasets. There are many other papers showing cDNA synthesis on fixed tissue that have not been cited, including FISSEQ (first step is in situ cDNA synthesis) and Crino et al, etc.

We further improved the citations by including two papers that reported in situ sequencing technologies, FISSEQ (Nguyen, et al., 2020. *Nat Methods*; Alon, et al., 2021. *Science*) in the INTRODUCTION and DISCUSSION sections. We could not find the paper by Crino et al. on spatial transcriptomics.

The authors response to the worry about small RNAs acting as primers is welcome, but unfortunately not sufficient. It is unfortunately still a problem as T7 RNA polymerase can cause nonspecific RNA amplification independent of T7 promoter function due to binding to nonspecific 3' overhangs on double stranded DNA (generated during the molecular biology steps) and acting as sites for nonpromoter initiated RNA synthesis. While it is true these would not have the UMI specifically introduced by cDNA synthesis, dependent upon the parameters for calling a sequencing read a real read (% error tolerated, etc.), a region could mistakenly be called a UMI. This is potentially a serious issue that would complicate the interpretation of these data.

As noted by the reviewer, it is widely known that T7 RNA polymerase occasionally initiates transcription independent of the T7 promoter. Despite this, most sequencing technologies based on in vitro transcription (IVT) amplification, including CEL-seq2, have successfully solved this problem and are now widely used (e.g., Ott, et al. 2017. *Nature*; Masuda, et al. 2019. *Nature*; and more). In CEL-seq2, the 2nd strand DNA synthesis produces double-stranded DNAs in the following order: T7

promoter, adapter, UMI, barcode, and cDNA. The antisense RNAs (aRNAs) are then transcribed by the IVT reaction, including an adapter sequence (Adapter 1; blue in the figure below) and the downstream sequences. The reverse transcription (RT) with the primers of another adapter (Adapter 2; green) produces cDNAs that have distinct adapters on both termini. These cDNAs are then conjugated with the P5 and P7 primers (Illumina) by PCR on both termini, thereby ensuring they hybridize to oligo DNAs on flow-cells for sequencing. Any aRNAs derived from T7 promoter-independent transcription lack the Adapter 1 sequence, so they are not conjugated with the P5 primer for sequencing. Considering that our PIC RNA-seq is based on CEL-seq2 technology in the library synthesis reactions, libraries from promoter-independent aRNAs would not be read also in our technology.

In activation of the oligonucleotide, Irradiation at 100 W at 100% power for 15 min. will clearly cause unwanted damage to RNA, DNA and tissue (there are several papers that highlight this including Klak et al., “Irradiation with 365nm and 405nm wavelength shows differences in DNA damage of swine pancreatic islets”). It will produce free radical formation (even in fixed tissue samples) causing crosslinking, oxidation, etc. This issue would be lessened if the uncaging time was reduced. Certainly for most such photoactivatable compounds the efficiency of uncaging should be milliseconds. The use of a 15min time window with 100% power is quite concerning.

In the Klak, et al. paper (2020. *PLoS One*), “living” cells were irradiated with 365 nm UVA to analyze DNA damage. However, it is unlikely that UVA will cause such damage in “fixed” cells because it is generally known that “direct” damage to DNA and RNA is more likely to be caused by the shorter ultraviolet wavelengths (UVB and UVC) than by UVA. Rather, “indirect” damage is caused by UVA through the enzymatic production of reactive oxygen species (ROS) such as the UVA-dependent activation of NADPH oxidase reported by Klak, et al. (2020. *PLoS One*). Although such an enzymatic effect is inconceivable in “fixed” cells, we performed experiments to evaluate possible direct and indirect damage by UVA.

To evaluate the direct damage of UVA to DNA and RNA, we designed a simple experimental system using polyadenylated synthetic RNAs with unique sequences (RNA1 and RNA2) or the mixture of 92 ERCC spike-in controls (see the figure above). These synthetic RNAs, in test tubes, were subjected to first strand synthesis with non-caged primers, then irradiated with UVA (365 nm) or UVC (254 nm) for 15 min. After 2nd strand DNA synthesis and the IVT reaction, a remarkable decrease in aRNAs was observed only in the UVC-irradiated samples. The subsequent sequencing analysis found no significant changes in the UVA-irradiated RNA samples with respect to their mapability and nucleobase substitution or deletion. The results are shown in the figure below.

To evaluate the indirect damage of UVA, we performed in situ RT on fixed NIH/3T3 cells and irradiated them with UVA (365 nm) for 15 min. The cell lysates were subjected to library preparation and sequencing according to the PIC RNA-seq protocol. We found that the UVA irradiation did not affect the yield of the libraries when non-caged primers were used for in situ RT. Read features also were similar with and without UVA irradiation. These data suggest that indirect damage by UVA is at negligible levels in fixed cells (see the figure below). Together, these results show that RNAs and DNAs are only marginally damaged by UVA irradiation in the conditions used for PIC RNA-seq.

Contrary to the authors response, loss of RNA upon heating may still be an issue as getting more amplified RNA doesn't necessarily mean less loss of RNA but as mentioned previously by the reviewer and the author, perhaps more RNA accessibility which would allow longer cDNA synthesis and longer amplified RNAs giving a higher yield, not necessarily showing little loss of cellular RNAs.

The qPCR for Gapdh does not address the reproducibility of the overall sample but rather a single RNA. In the described control experiment, It is difficult to envision how RNA from a fixed tissue section will yield more signal (Gapdh PCR signal was "comparable to, or slightly higher") than cell isolated RNA assessed by traditional CEL-seq2.

The 65°C heating step is generally performed before in situ and in vitro RT reactions, including CEL-seq2. To address the possible *loss of cellular RNAs* by the pre-heating treatment, we performed an additional experiment in which fixed NIH/3T3 cells were or were not heated before performing in situ RT and PIC RNA-seq according to the standard protocols. As expected,

the number of detected genes was increased by 65°C heating, clearly supporting our argument that there was *little loss of cellular RNAs*.

The reason why in situ RT provided more cDNAs than CEL-seq2 (Fig. 1g in the previous manuscript) would be explained simply by considering the thermodynamics of RT primers and mRNAs. In in vitro RT reactions, primers and mRNAs are made freely mobile by thermal fluctuation, so the hybridization between poly-T and poly-A sequences can be easily dissociated. In in situ RT reactions however, mRNAs are immobilized to the cells, so the A:T duplex with RT primers will be robust to the thermal fluctuation, resulting in a higher yield of cDNAs than in vitro RT reactions. Decreasing the free energy of DNAs and RNAs is known to elevate T_m values as exemplified by locked nucleic acids in which the random motion of DNA or RNA strands is locked by bridging between the 2'-oxygen and 4'-carbon atoms of the ribose moiety, a technique that is now widely used to enhance hybridization, RNA interference, and RT reactions (Hagedorn, et al. 2018. *Drug Discov Today*). We decided to remove Fig. 1g that was in the previous manuscript because the data in the figure were not essential for developing the PIC RNA-seq protocol and might confuse a reader.

The additional data showing 10,000 genes in SG and nuclear speckles by PIC is also troubling. As the site of precursor RNA processing, nuclear speckles will contain 10-100x less RNA than the cytoplasm where RNA stability allows RNA abundances to be high (hnRNA is always significantly less than cytoplasmic RNA). Further, technically it should be much harder to detect RNAs in nuclear speckles (given their lower amounts) and hence there should technically (not just biologically) be fewer genes identified than from cytoplasmic RNA. Why there would be a DEG of 1990 genes for NS-positive vs

NS-negative samples is also curious as they should be very similar in composition as the NS is the processing site for cytoplasmic RNAs. But as highlighted earlier the detectability will be less as there should be less RNA detected in NS than the cytoplasm.

There are a few misunderstandings. First, we did not compare RNAs in NSs and the cytoplasm, as described in the previous and current manuscripts. We photo-irradiated NS-positive regions (NS-posi) and prepared the control samples where NS-negative regions in the nucleus (NS-nega) and the entire nucleus (both) were photo-irradiated. Then, DEGs were detected by comparison between NS-posi and NS-nega samples, but not between NS-posi and cytoplasm-irradiated samples.

Second, it is not surprising to find many RNA species in NSs, because a number of papers have shown that poly-A RNA is abundantly localized in NSs (e.g., Wang, et al. 2018. *J Cell Biol*; Narcís, JO., et al. 2018. *Sci Rep*; Hall, LL., et al. 2006. *Anat Rec A Discov Mol Cell Evol*; and more). To confirm that this was the case under our culture conditions, we performed poly-T FISH and detected strong signals in NSs (see below). In addition, we performed in situ RT with non-caged primers and fluorescent nucleotides and detected abundant cDNAs in NSs (see below), indicating that NSs in cells we used in the manuscript contain poly-A RNA at detectable levels by in situ RT. However, the absolute amount of poly-A RNA in NS of single cell was expected to be low due to the small size of NSs and so we used 100 cells for PIC RNA-seq library. Based on the statistical analysis of five replicates, we detected large number of DEGs. We believe our data was rigorously designed and conducted, considering the abundance of RNA in NSs, and the number of cells and replicates used for library preparation.

Finally, it is a minor point but the discussion of differences in rebuttal to reviewer 1, between PIC and LCM are misleading as they both rely upon laser characterization of the area to be transcriptionally assessed and as long as the area can be visualized there should be little loss due to complicated morphologies.

Amongst the LCM-coupled transcriptomics studies, the highest spatial resolution was demonstrated by LCM-seq, succeeding single-cell RNA-seq (Nichterwitz, et al. 2016. *Nat Commun*; see the figure, scale bar is 50 μm). However, because the thickness of the cutting edge is several micrometers (d in the figure), it may be impossible to pick out subcellular components (e.g., stress granules and NSs) and cells with complicated shapes (e.g., neurons and astrocytes).

Reviewer #3 (Remarks to the Author):

The authors have answered the comments from reviewer 3 in an appropriate manner. The addition of single cell, 10 cell etc data to the manuscript is a great improvement. The technology displays a high sensitivity, capturing a lot of UMI and genes, not seen in similar methods, as the authors point out. The title has been updated to include "High-depth" which does not sound grammatically sound to me. Perhaps "High sensitivity" is better, but a native English speaker would know best.

We thank the reviewer for the comment. "High-depth" means that a larger number of per-area transcripts can be detected from a small region by PIC RNA-seq than existing spatial transcriptome technologies, which was checked again and endorsed by a native English speaker.

REFERENCES

- Ding, J., et al. Systematic comparison of single-cell and single-nucleus RNA-sequencing methods. *Nat Biotechnol.* 2020. 38(6):737-746.
- Mereu, E., et al. Benchmarking single-cell RNA-sequencing protocols for cell atlas projects. *Nat Biotechnol.* 2020. 38(6):747-755.
- Hayashi, T., et al. Single-cell full-length total RNA sequencing uncovers dynamics of recursive splicing and enhancer RNAs. *Nat Commun.* 2018. 9(1):619.
- Nguyen, HQ., et al., 3D mapping and accelerated super-resolution imaging of the human genome using in situ sequencing. *Nat Methods.* 2020. 17(8):822-832.
- Alon, S., et al. Expansion sequencing: Spatially precise in situ transcriptomics in intact biological

systems. *Science*. 2021. 371(6528):eaax2656.

Ott, PA., et al. An immunogenic personal neoantigen vaccine for patients with melanoma. *Nature*. 2017. 547(7662):217-221.

Masuda, T. et al. Spatial and temporal heterogeneity of mouse and human microglia at single-cell resolution. *Nature*. 2019. 566(7744):388-392.

Klak, M., et al. Irradiation with 365 nm and 405 nm wavelength shows differences in DNA damage of swine pancreatic islets. *PLoS One*. 2020. 15(6):e0235052.

Valencia, A., et al. Nox1-based NADPH oxidase is the major source of UVA-induced reactive oxygen species in human keratinocytes. *J Invest Dermatol*. 2008. 128(1):214-22.

Hagedorn, PH., et al. Locked nucleic acid: modality, diversity, and drug discovery. *Drug Discov Today*. 2018. 23(1):101-114.

Wang, K., et al. Intronless mRNAs transit through nuclear speckles to gain export competence. *J Cell Biol*. 2018. 217(11):3912-3929.

Narcís, JO., et al. Accumulation of poly(A) RNA in nuclear granules enriched in Sam68 in motor neurons from the SMN Δ 7 mouse model of SMA. *Sci Rep*. 2018. 8:9646.

Hall, LL., et al. Molecular anatomy of a speckle. *Anat Rec A Discov Mol Cell Evol Biol*. 2006. 288(7):664-75.

Nichterwitz, S., et al. Laser capture microscopy coupled with Smart-seq2 for precise spatial transcriptomic profiling. *Nat Commun*. 2016. 7, 12139

Reviewers' Comments:

Reviewer #4:

Remarks to the Author:

While I find that the majority of reviewer 2's comments have been addressed by the authors and commend the authors for performing new experiments to address comments on DNA damage, I agree with some of reviewer's concerns on the tissue profiling and nuclear speck profiling sections of the manuscript. In both sections, there is no systematic/quantitative validation of PIC RNA-seq findings which makes it more challenging to assess the sensitivity and accuracy of the method.

1) In the neural plate profiling experiments, while it is nice to see some ISH validation, the lack of DEGs between VM and MM and the lower number of DEGs between VM-DL compared to MM-DL were concerning. Are there any existing "ground truth" datasets available, such as scRNAseq, micro-dissected regional neural plate bulk RNA-seq or large scale ISH, that can be used to quantify the percentage of known regional expression differences between these areas that are successfully captured by PIC-seq? Frankly it would have been more useful to see an application to a tissue that is well characterised spatially with known cell type patterns, such as the mouse brain where many other spatial txn methods have been tested on.

2) The NS/SG experiments are interesting and I understand that there may not be a good ground truth dataset out there. However, it would have been more convincing to see smFISH validation of some of the new NS/SG genes. Only one gene is mentioned to be known NS gene, Malat1. If there are no other known genes, some smFISH validation would strongly improve the strength of these findings.

To add, the sensitivity comparison between PIC and other ST methods based on the NS/SG profiling of cells (i.e. number of UMI in micron square profiled area) is somewhat misleading. Other methods have profiled tissue material, while this study is looking at cultured cells. To present a more balanced comparison, the authors should compare their tissue results to other methods or remove this statement.

Reviewer #4 (Remarks to the Author):

While I find that the majority of reviewer 2's comments have been addressed by the authors and commend the authors for performing new experiments to address comments on DNA damage, I agree with some of reviewer's concerns on the tissue profiling and nuclear speck profiling sections of the manuscript. In both sections, there is no systematic/quantitative validation of PIC RNA-seq findings which makes it more challenging to assess the sensitivity and accuracy of the method.

1) In the neural plate profiling experiments, while it is nice to see some ISH validation, the lack of DEGs between VM and MM and the lower number of DEGs between VM-DL compared to MM-DL were concerning. Are there any existing "ground truth" datasets available, such as scRNAseq, micro-dissected regional neural plate bulk RNA-seq or large scale ISH, that can be used to quantify the percentage of known regional expression differences between these areas that are successfully captured by PIC-seq? Frankly it would have been more useful to see an application to a tissue that is well characterised spatially with known cell type patterns, such as the mouse brain where many other spatial txn methods have been tested on.

We have carefully considered the reviewer's insightful comments. Accordingly, we performed PIC RNA-seq for hippocampal regions (CA1, CA3, and dentate gyrus [DG]) of adult mouse brains. The PIC analysis demonstrated that the expression profiles were clearly separated into three groups by principal component analysis according to the photo-irradiated sites. DEGs were detected between pairs of groups, with 231, 717, and 672 genes being statistically significantly differentially expressed in the CA1 vs CA3, CA1 vs DG, and CA3 vs DG comparisons, respectively. These DEGs included the genes validated by Cembrowski et al. with ISH (Cembrowski, et al., 2016. *eLife*), such as those specifically expressed in CA1 (*Wfs1*), CA1 and CA3 (*Ociad2* and *Dkk3*), and DG (*Prox1* and *Pdzd2*). We have added these data to Fig. 3 and the corresponding RESULT section, and moved the results of PIC in the neural tubes to Supplementary Fig. 2.

2) The NS/SG experiments are interesting and I understand that there may not be a good ground truth dataset out there. However, it would have been more convincing to see smFISH validation of some of the new NS/SG genes. Only one gene is mentioned to be known NS gene, Malat1. If there are no other known genes, some smFISH validation would strongly improve the strength of these findings.

In addition to *MALAT1*, we performed smFISH for several other genes that were detected as NS-specific DEGs and found that the *AKRIC2* transcripts were localized in, or at the periphery of, NSs (Fig. 5i). We thank the reviewer for providing us with critical comments on this experiment.

To add, the sensitivity comparison between PIC and other ST methods based on the NS/SG profiling

of cells (i.e. number of UMI in micron square profiled area) is somewhat misleading. Other methods have profiled tissue material, while this study is looking at cultured cells. To present a more balanced comparison, the authors should compare their tissue results to other methods or remove this statement. We evaluated the number of per-area UMIs for the PIC data in the hippocampus. We found that the numbers of gene-assigned UMIs relative to the total irradiated areas were 50.3, 28.5, and 80.7 UMIs/ μm^2 for CA1, CA3, and DG, respectively (Fig. 4d). We removed the sentences describing the comparison with other spatial transcriptome methods because the method of estimating the number of per-area transcripts is technically different among PIC (UMIs per photo-irradiated area), seqFISH (FISH foci per observed area), and Slide-seq and HDST (UMIs per size of a barcoded bead).

Reviewers' Comments:

Reviewer #4:

Remarks to the Author:

The authors have addressed my comments with new experiments profiling the hippocampus and additional smFISH validation. While they observe over 1000 DEGs between different hippocampal regions, one might expect to see more DEGs between these regions based on scRNAseq findings. However, it might not be possible to make a direct comparison as the bulk regions profiled in this paper each contain multiple cell types. It would have been interesting to see a comparison to Visium here, based on publicly available datasets, but I do not find this necessary for publication. Given the authors' revisions, I would recommend for publication.

Reviewer #4 (Remarks to the Author):

The authors have addressed my comments with new experiments profiling the hippocampus and additional smFISH validation. While they observe over 1000 DEGs between different hippocampal regions, one might expect to see more DEGs between these regions based on scRNAseq findings. However, it might not be possible to make a direct comparison as the bulk regions profiled in this paper each contain multiple cell types. It would have been interesting to see a comparison to Visium here, based on publicly available datasets, but I do not find this necessary for publication. Given the authors' revisions, I would recommend for publication.

We thank the reviewers for their constructive comments on PIC RNA-seq of mouse brain. We agree that it would be useful to compare PIC with other spatial transcriptome technologies, including single-cell RNA-seq and the Visium platform, although he/she does not request to be included in this study. In light of this, we have added the following sentence to the DISCUSSION:
“For the further development of spatial transcriptomics, it will be useful in the future to benchmark and compare various techniques including PIC at the single cell level.”